# Antibiotic Resistance Diagnosis in ESKAPE Pathogens—A Review on Proteomic Perspective

**DOI:** 10.3390/diagnostics13061014

**Published:** 2023-03-07

**Authors:** Sriram Kalpana, Wan-Ying Lin, Yu-Chiang Wang, Yiwen Fu, Amrutha Lakshmi, Hsin-Yao Wang

**Affiliations:** 1Department of Laboratory Medicine, Linkou Chang Gung Memorial Hospital, Taoyuan 333423, Taiwan; 2Syu Kang Sport Clinic, Taipei 112053, Taiwan; 3Department of Medicine, Harvard Medical School, Boston, MA 02115, USA; 4Department of Medicine, Brigham and Women’s Hospital, Boston, MA 02115, USA; 5Department of Medicine, Kaiser Permanente Santa Clara Medical Center, Santa Clara, CA 95051, USA; 6Department of Biochemistry, University of Madras, Guindy Campus, Chennai 600025, India

**Keywords:** antibiotic resistance, bacterial resistance, ESKAPE, MALDI-TOF, mass spectrometry, proteomics

## Abstract

Antibiotic resistance has emerged as an imminent pandemic. Rapid diagnostic assays distinguish bacterial infections from other diseases and aid antimicrobial stewardship, therapy optimization, and epidemiological surveillance. Traditional methods typically have longer turn-around times for definitive results. On the other hand, proteomic studies have progressed constantly and improved both in qualitative and quantitative analysis. With a wide range of data sets made available in the public domain, the ability to interpret the data has considerably reduced the error rates. This review gives an insight on state-of-the-art proteomic techniques in diagnosing antibiotic resistance in ESKAPE pathogens with a future outlook for evading the “imminent pandemic”.

## 1. Introduction

Bacterial antimicrobial resistance (AMR) has emerged as a leading public health hazard, and the World Health Organization (WHO) has acknowledged an estimated 10 million people being killed annually by 2050 [1]. A synchronized action plan is needed to address the “imminent pandemic” of AMR. The US Centers for Disease Control and Prevention (CDC) reported 18 AMR threats, and the European Union and the European Economic Area reported 16 AMR threats in the estimated burden of eight pathogens. The foremost challenge is the multidrug resistance burden of six pathogens [2] prioritized by the WHO, namely *Escherichia coli*, *Staphylococcus aureus*, *Klebsiella pneumoniae*, *Acinetobacter baumannii*, *Pseudomonas aeruginosa*, and *Enterobacter* (ESKAPE), that contribute to the burden of AMR [3]. The ESKAPE pathogens are also listed by the Global Action Plan on AMR, the UN Interagency Coordination Group, and the One Health Global Leaders Group [4]. More attention is needed in funding research and development in understanding the drug resistance in each of the ESKAPE pathogens. Limitations in developing new and effective antibiotic treatments arise from the lack of a coordinated global assessment of bacterial AMR.

Bacterial AMR burden increases with increased antibiotic usage in high-resource settings and is a function of bacterial resistance and acute infections. Other factors include the lack of microbiological testing to prevent inappropriate antibiotic use, inadequate guidelines, and the easy procurement of antibiotics [5]. The AMR burden increases primarily from the inadequate availability of second- and third-line antibiotics, the availability of fake or inferior antibiotic drugs [6,7,8], and the lack of sanitation and hygiene [9,10,11]. The AMR pattern varies globally, with the existence of pathogen variants and variations in pathogen–drug interactions. Antibiotic stewardship is an integrated approach in controlling the spread of AMR. Limiting access to antibiotics is not viable in life-saving situations particularly when second-line antibiotics are unavailable [7].

Several innovative approaches are available to improve antibiotic use [12]. Logically the most effective way would be to allow the choice of narrow-spectrum antibiotics, reducing the intensity of the selection of a broad spectrum of antibiotics and reserving them for situations that are in real need which can be accomplished by the early detection of bacterial susceptibility [13]. Resistance diagnosis enables the “search and destroy” tactics to combat potentially dangerous pathogen strains [14,15,16]. A targeted infection control measure such as “search and destroy” may be possible from a rapid and accurate identification. This review comprehensively describes the proteomic profile of resistance diagnosis with a special focus on ESKAPE pathogens AST including the state-of-the-art techniques available, their roles ranging from typing to drug selection, and their advantages despite their limitations.

## 2. Antibiotic Resistance and Resilience

Antibiotic resistance is the clinical stance resulting from the insensitivity to an antibiotic drug and is categorized in terms of the minimum inhibitory concentration (MIC) in μg/mL, which is the lowest concentration of the drug that inhibits bacterial growth [17]. Microorganisms with an MIC beyond the normal distribution are resistant. According to EUCAST, a Susceptible (S) organism has therapeutic success with a standard dosing regimen, a Susceptible (I) organism has therapeutic success requiring increased exposure, and a Resistant (R) organism has therapeutic failure even with increased exposure.

Antibiotic resilience is the ability of the organism to recover from antibiotic stress and is expressed as the minimum duration for killing (MDK). By definition, it is the time needed for 50% of the total biomass to recover after antibiotic treatment. Resilience is described by a range of aspects such as bacterial tolerance, persistence, recalcitrance, adaptation, etc. These aspects reflect the diverse machinery prevailing in the bacteria to withstand antibiotic treatments and any of the related disturbances [18]. Identifying the determinants of bacterial resistance and resilience is crucial for understanding the response and strategy development. For instance, the resistance–resilience analysis framework has helped in the identification of the phenotypic signatures of extended-spectrum β-lactamase (ESBL) bacteria and has been used as a guide in combination treatments [19].

## 3. Methods of Resistance Profiling and Diagnosis

Many mechanisms are involved in bacterial antibiotic resistance [20]. Antibiotic resistance is exhibited on a genetic or mechanistic basis. Genetic basis includes mutational resistance that modifies antimicrobial targets such as decreasing drug uptake, activating drug efflux, or modulating regulatory networks. Mutations in genes arise from the acquisition of antibiotic resistance genes by horizontal gene transfer by a variety of mechanisms chiefly involving conjugation in which mobile genetic elements (MGEs) notably plasmids and transposons are mobilized. MGEs are crucial in the development and dissemination of antimicrobial resistance among clinically relevant organisms.

Mechanistically, bacteria acquire resistance by modifications of the antibiotic (chemical alteration or destruction), decreased penetration and efflux of the antibiotic, change in the target site (modification/mutation/bypass), and resistance due to global cell adaptations. Resistance is also induced by virulence factors such as biofilm formation associated with desiccation [21,22].

Profiling microbial resistance has come a long way. Primarily, microbial-culture-based methods identify the bacterial phenotype, whereas bacterial strain and species identification requires biochemical assays [23]. The evaluation of antimicrobial resistance based on the disk diffusion or broth dilution methods quantitatively evaluates resistance in terms of MICs [24] based on the protocols prescribed by the Clinical and Laboratory Standards Institute (CLSI) and the European Committee on Antimicrobial Susceptibility Testing (EUCAST) as the “gold standard” [25]. These growth-based tests have limitations of longer turn-around times of 12 to 72 h [25,26]. The bacterial cultivation adds further 18 to 24 h for biochemical characterization [27]. The disadvantages of culturing methods include collection conditions and specific growth media requirements that generate errors leading to a lack of sensitivity [28,29]. In addition, routine bacterial culture methods are not practically applicable to non-cultivable pathogens [30,31]. Automated AST systems such as the Vitek 2, Phoenix, and MicroScan WalkAway perform in a simplified workflow and reduced time to results compared to traditional methods but still need bacterial culturing [32]. Therefore, the highly specific and sensitive molecule-based approach has also been used in the bacterial identification of resistance [33].

Molecular-based approaches amplify or hybridize genetic sequences encoding specific resistance determinants using conventional polymerase chain reactions (PCRs), quantitative real-time polymerase chain reactions (RT-PCRs), or DNA-micro-arrays with more sensitivity, specificity, and shorter turn-around times [23,34,35] which is not possible in culture-based methods [36]. However, these methods require detectable levels of DNA in conditions of low-abundance genes and heteroresistance [25]. Culture-independent methods are limited with variable clinical sensitivities [37]. Digital PCR systems detect low-abundance targets and heteroresistance analysis immediately without requiring prior culture enrichment [38]. Pitfalls with the molecular-based techniques are results with false-positive outcomes due to the amplification of silent genes or pseudogenes and false-negative outcomes due to mutations in the primer binding sites. Conventional PCRs also fail to detect hypervariable organisms and rapidly changing mechanisms, particularly in the detection of Gram-negative bacteria, including ESBL strains and carbapenem-resistant *Enterobacteriaceae* (CREs) with single-nucleotide polymorphisms (SNPs) [39]. Certain resistance markers detected by the PCR do not correlate with phenotypic resistance [40].

Culture-independent techniques termed nucleic acid testing (NAT) perform quicker diagnosis with higher sensitivity [41] but require prior information on the pathogen under test and its nucleic acid sequences [26]. The methods include molecular methods such as the PCR, RT-PCR, loop-mediated isothermal amplification (LAMP), nucleic acid sequence-based amplification (NASNBA), transcription-mediated amplification (TMA), and strand displacement amplification (SDA) [42]. Highly multiplexed PCR panels simultaneously detect bacterial pathogens that commonly cause specific clinical syndromes [37]. The other non-targeted methods that do not require prior nucleic acid information include next-generation sequencing (NGS) technologies [43,44] which in combination with bioinformatics provide accurate detection and characterization of pathogens and predict the strains evading vaccines [36]. Metagenomics NGS (mNGS) provides accurate data on the composition of microbial communities that are impossible to culture [45].

Clinical microbiology has also focused on applying genomics for AST. Whole-genome sequencing (WGS) is the primary genomic approach that predicts strains of all prevalent resistant phenotypes accurately and consistently. It ascertains the simultaneous identification of antibiotic-resistant phenotypes from the entire genome by screening multiple loci. The data from the genome sequence are stored digitally and are independent of primer specificity reducing false-negative results [46]. With a huge availability of data in the public domain, antimicrobial resistance determinants are readily identifiable with both the whole-genome and NGS technologies [39]. As a primer-independent method, it detects antibiotic resistance rapidly but is capable of detecting only previously documented mechanisms [47,48].

Around the mid-2000s, innovations in sequencing technology helped develop second-generation instruments based on Illumina sequencing that provided short-sequence (≤300 bp) reads and paired ends at reduced costs. From about 2010, third-generation sequencing innovations helped develop Oxford Nanopore Technologies (ONT) and Pacific Biosciences (PacBio) technologies that produce longer reads > 2 Mb [13] with fewer gaps allowing tandem repeats and nested insertions [49], but they have higher error rates than Illumina [50]. Hybrid assemblies achieve accurate results by combining the accuracy of Illumina as well as using the longer reads of ONT/PacBio to overcome the shortcomings of both technologies. As new approaches emerge, clinical sequencing constantly shifts for cost effectiveness. The minimum cost is around 80 USD/genome which is expensive for routine use in clinical laboratories, although results are obtained within less than 24 h. Although the ONT Flongle disposable flow cells are less expensive, sequencing problems prevail in prediction as false positives are introduced from sequencing errors and DNA contamination from other organisms [46].

## 4. Proteomic Tools in Antibiotic Resistance

Proteomic analysis provides more functionally or clinically relevant information than genetic/genomic testing as protein levels indicate the actual functional status of the cell. Proteomic studies are capable of analyzing the protein—expression, post-translational modifications, and turnover rates [51]. Bacterial proteomics utilize both gel-based and non-gel-based techniques. Preliminary analysis involves a two-dimensional gel electrophoresis (2-DE) followed by the analysis of the gel image. The differential expression of protein is analyzed by a differential-in-gel electrophoresis (DIGE) technique using a fluorescent dye [52]. With the advancement of technologies, mass spectrometric analysis directly quantifies the protein as well as its functional status [53]. The bacterial proteomics have emerged on par with the proteomic tools developed (Figure 1).

Mass spectrometry (MS) analyzes ionized samples separated on the basis of the mass-to-charge (*m*/*z*) ratio and detects as a mass spectrum. Ionization techniques namely electrospray (ESI) and matrix-assisted laser desorption/ionization (MALDI) ionize analytes in a solution and a dry, crystalline matrix, respectively. Mass analyzers are of four types: ion trap, time-of-flight (TOF), quadrupole, and Fourier transform ion cyclotron (FT-MS). Time-of-flight (TOF) instruments analyze complex peptide mixtures [54], with a high acquisition rate permitting coupling with ion mobility spectrometry (IMS) [55]. In IMS, ion separation is size- and shape-based or based on the collisional cross section (CCS) [56]. TOF analyzes the emerging ions in a ms or sub-ms time frame. IMS nested between LC and MS or any additional dimension of separation is referred to as IMS-MS [57,58]; it increases the speed of analysis and selectivity [59] in highly complex proteomics samples and adds to the fourth dimension of proteomic analysis [60,61].

Relative quantification in MS is of two types: label and label-free methods. Label-free quantification uses MS signal intensity or spectral counting which is directly proportional to the peptide concentration. In the label method, the protein/peptide is labeled with a stable isotope tag chemically, metabolically, or enzymatically. Stable isotopes exhibit slight mass differences compared to their unlabeled counterparts that produce distinguishable signals in MS. In vivo protein is labeled by growing the cells in isotopically labeled amino acids—13C or 15N—referred to as the stable isotope labeling by amino acids in cell culture (SILAC) method [62].

Proteins/peptides are also tagged chemically with isobaric mass tags known as tandem mass tags (TMTs), which on fragmentation yield reporter ions of differing mass that is quantified. The isobaric tags for relative and absolute quantitation (iTRAQ) method quantitates the relative protein levels from different sources in a single experiment. The workflow is that the isobaric label is covalently attached to peptides after protein digestion, and samples are pooled, fractionated by LC, and analyzed in a tandem mass spectrometer (MS/MS). The relative quantification of the protein is obtained from the combined ratios of proteins/peptides. iTRAQ combined with MALDI or ESI-MS/MS provides accurate information on the relative protein concentration.

In the absolute quantification (AQUA) method, internal standard peptides are labeled and added to the sample during proteolytic digestion. The relative protein concentration is quantitated from both isotope-labeled AQUA peptides and unlabeled native peptides and measured by selected reaction monitoring (LC-SRM). From the known amount of the internal standard, the MS determines the ratio between the internal standard and analyte [63]. It provides the relative quantity of protein and information on the post-translational modifications.

In trapped ion mobility spectrometry (TIMS), the ions are rested in an ion tunnel device and balanced in a stream of gas at a low electrical potential [64]. The time-resolved ions are released into the mass analyzer downstream [65]. Another MS-based approach is parallel accumulation–serial fragmentation (PASEF) [66] which synchronizes the MS/MS precursor selection with TIMS separation. TIMS scan acquires more than one precursor, and peptides are continually selected for sequencing. Typical DDA measurements are performed after a survey scan, and the N-highest abundant precursor ions are targeted for MS/MS analysis [67]. The MS/MS spectrum quality is improved from the fast acquisition speed (50–200 ms for a full scan) and the repeated re-targeting of low-abundance precursors.

In the bio-orthogonal noncanonical amino acid tagging (BONCAT) method, noncanonical amino acid (ncAA) is incorporated into protein, conjugated to an affinity tag, and enriched. The enriched proteins are identified and quantified by LC-MS/MS [68,69]. Its advantage is that labeled proteins are separated physically from the remaining proteome [70]. The ncAA pulse time is only a few minutes in bacteria and thus quantifies dynamic processes. Extended pulse times identify proteins synthesized at extremely low rates under anaerobic conditions such as in the survival of *Pseudomonas aeruginosa* [71]. ncAA incorporation uses a mutant aminoacyl-tRNA synthetase and is expressed according to the target cell [72] or specific cell state condition [73].

Raman spectroscopy is an optical technique, where the sample is irradiated, and the scattered light is analyzed. The shift between the frequency of incident and scattered light is the Raman effect. This shift is induced by molecular vibrations in the sample which are distinct for a bacterial cell based on protein, lipids, and DNA and are the “chemical fingerprint”. Based on Raman spectroscopy, a single-cell Raman spectrum (SCRS) is used in the identification of microbes at a single-cell level as it provides a spatial resolution of <1 μm^3^. Like FACS, a laser beam is applied on a single cell, and the Raman spectrum is obtained. On the basis of a shift in the spectra, cell type and phenotypic changes in bacteria are characterized [74,75]. When labeling with stable isotopes such as 13C, 15N, and 2 H(D), bacteria also display a characteristic Raman spectrum shift due to heavy isotopic atom replacement with biomolecules. For instance, labeling with D_2_O incorporates D into biomolecules forming carbon-deuterium (C-D) bonds that show a distinguishable Raman band (2000–2300 cm^−1^) shifted from C-H vibration [76].

Raman-based bacterial identification distinguishes bacteria susceptibility from the absence of a Raman band in resistant bacteria which is referred to as an “antibiotic effect signature” [77,78,79,80]. Employing the same principle, the fast Raman-assisted antibiotic susceptibility test (FRAST) method was developed. The clinical protocol includes steps of Raman-based single-cell GS classification, two-step antibiotic inhibition, D_2_O labeling, SCRS acquisition, and data analysis [81,82,83] from subtle variations in SCRS.

Raman spectroscopy distinguishes bacterial strains [84,85,86]. Surface-enhanced Raman spectroscopy (SERS) analysis of bacteria discriminates Gram-positive and Gram-negative strains, and the classification is based on linear discriminant analysis (LDA) [87]. Classical Gram staining requires 16–24 h, delaying the rapid diagnosis. However, in the FRAST, Gram classification is obtained within a few hours, and its accuracy reaches 100% when >90% of single cells have been Gram classified. The dual-mode detection—Gram classification antibiotic susceptibly detection by the FRAST—makes a stand-alone automatic Raman infection diagnostic system possible.

In another latest development, the direct-on-target microdroplet growth assay (DOT-MGA) measures bacterial growth on the MALDI-TOF MS directly by incubating with and without the indicator antibiotic in the microdroplet nutrient broth. It is based on the principle of the broth microdilution method, with a modification in which the bacteria are incubated on the MS target plate, and bacterial growth is determined at the breakpoint concentration based on MS identification scores. By assessing the growth in the presence of various antibiotics, the potential sensitivity mechanisms of drug resistance are analyzed. It is superior to the broth microdilution method and the direct-from-blood-culture disk-diffusion method in terms of speed and easy operation. Nix et al., 2020 employed the DOT-MGA in the rapid detection of pathogens in the blood culture of methicillin-resistant *Staphylococcus aureus* (MRSA) patients [88] producing reliable results within 4 h incubation in determining carbapenemase resistance in *K. pneumoniae*, *E. cloacae*, *E. aerogenes*, *P. mirabilis*, and *K. aerogenes* [89].

The emergence of bacterial proteomics on par with the proteomic technologies is illustrated in Figure 1.

### Top-Down versus Bottom-Up Proteomics

Proteomics has evolved from simple gel-based (2-DE or 1-DE gel-LC-MS/MS) to gel-free methods. Proteomic analysis is categorized as “top-down” and “bottom-up”. “Top-down” proteomic analysis encompasses intact proteins, whereas “bottom-up” analyses involving proteolytically digested protein are categorized into three types: “shotgun” or untargeted proteomics that is MS-operated in a data-dependent acquisition (DDA) mode; targeted proteomics carried out by multiple reaction monitoring (MRM); and the data-independent acquisition (DIA) method [90].

In top-down proteomics, intact protein analysis enables the analysis of protein isoforms and the stoichiometry of post-translational modifications (PTM). In the bottom-up approach, digested proteins are separated by LC and ionized and mass-analyzed with a mass spectrometer in full scans (MS), and fragments are selected in N consecutive MS/MS scans. Targeted proteomics detect and quantify a predetermined set of peptides by selected reaction monitoring (SRM) which is multiple reaction monitoring (MRM) from a set comprehensive protein database [62].

The “data-independent acquisition” (DIA) method is unbiased and involves cyclic recording in the entire LC time range and the fragment ion spectra contained in predetermined isolation windows. A combination of the DIA method with a targeted data extraction strategy is the “sequential isolation window acquisition theoretical mass spectra” (SWATH MS) in which the user-defined *m*/*z* window is fragmented and correlated to previously generated query parameters and scored [91].

The advantages of a gel-free/label-free proteomic technique with the potential application of proteomics in bacterial pathogen studies including comparative proteomics and differential protein expression in response to antibiotic treatment clearly explain the superiority of proteomic technologies (Figure 2).

## 5. ESKAPE Pathogens

Recently, an extensive review described the antibiotic resistance mechanisms identified in pathogens given priority status, i.e., ESKAPE (*Enterococcus* spp., *Staphylococcus aureus*, *Klebsiella pneumoniae*, *Acinetobacter baumannii*, *Pseudomonas aeruginosa*, and *Enterobacter* spp.) [92]. In ESKAPE pathogens, resistance develops through genetic mutations and the acquisition of MGEs [93]. ESKAPE pathogens are resistant to oxazolidinones, lipopeptides, macrolides, fluoroquinolones, tetracyclines, β-lactams, β-lactam–β-lactamase inhibitor combinations, and last-line antibiotics including carbapenems, glycopeptides, and polymyxins [94]. Therefore, preclinical and clinical trials encompassed many treatment options including vaccine development to control the burden. Unfortunately, no vaccines are available for ESKAPE infections [5].

Both in clinical settings and at the community level, ESKAPE pathogens serve as the model organism for resistance. Despite their heterogeneity, the overall mechanisms involved in the emergence and persistence are shared by all ESKAPE pathogens individually. ESKAPE pathogens are highly prevalent in the clinical setting due to their ability to form a biofilm on abiotic and biotic surfaces. Apart from drug development, inappropriate use of antibiotics, and sustained stewardship, improved diagnosis is essential to control ESKAPE AMR burden.

### One Health Approach in ESKAPE Management

In 2017, the European Union implemented the “One Health” approach to combat antibiotic resistance [95] recognizing the need for safeguarding human health by protecting animal and environmental health as well as related fields [96]. ESKAPE bacteria with AMR are widely distributed into the environment and ecosystem [97]. Among the ESKAPE pathogens, *Pseudomonas* and *Acinetobacter* are enteric bacteria and soil commensals that are ubiquitous in livestock animals and slaughterhouse wastewater discharges. Outbreaks in veterinary hospitals are relevant to the isolation of MRSA, vancomycin-resistant *Enterococcus* (VRE), and ESBL, producing *Escherichia coli*, *Klebsiella pneumoniae*, and *Acinetobacter baumannii* strains from humans, livestock, and contaminated food [98]. Most ESKAPE isolates are multidrug-resistant (MDR) with the highest risk of mortality. The consequences of AMR-related infections are related to iatrogenic disease states in which treatment of the infection results in co-morbidities [99]. Therefore, the key aspect of the One Health concept may be addressed from the early and rapid diagnosis of resistance.

## 6. Proteomic Studies on ESKAPE Resistance

### 6.1. Enterococcus spp.

*Enterococci* are Gram-positive cocci, facultatively anaerobic, inhabiting the gastrointestinal tract, and cause a variety of infections including urinary tract infections, bacteremia, intra-abdominal infections, and endocarditis [100]. *Enterococci* develop antibiotic resistance both intrinsically and by acquisition. They develop resistance to cephalosporins, aminoglycosides, lincosamides, and streptogramins intrinsically [101] and thus acquire added resistance from MGEs [102]. The malleability of genomes and the disseminating determinants are attributed to their adaptation to harsh environments. Thus, both the microbial and host factors convert the second-rate pathogen into a first-rate clinical problem [103].

Aminoglycoside resistance in *Enterococci* is acquired from the plasmid-borne resistance factor [104] and also by aminoglycoside-modifying enzymes from the mobile elements [105]. β-lactamase activity in *Enterococci* compromises combination therapy [106]. A comprehensive proteomic study on vancomycin-resistant the *Enterococci faecalis* strain revealed the proteins vital for antibiotic resistance [107]. It included the pheromone-binding proteins involved in the conjugative plasmid transfer [108], the detection of which could aid in antibiotic cross selection. An LC-ESI/MS-based proteomic study predicted the functions of pheromone precursors, pheromone/peptide-binding components of ABC transporters, and basic membrane proteins [109].

MALDI-TOF MS spectra with artificial intelligence (AI) discriminated the proteomic patterns of VRE from the vancomycin-susceptible *Enterococci faecium* (VSE) [110,111]. The proteomic profile of VRE revealed the elongation factor EF-Tu in the cytoplasm and elongation factor G (EF-G) in the membrane [107] that moonlights the link with target receptors from host cell membranes and paves the way for colonization [112]. These data predict the timing of the introduction of prodrugs affecting Ef-Tu and regulating bacterial elongation.

Of the several stress factors in *Enterococci faecalis* [113], the antiphagocytic factor Cold shock protein A (CspA) encoded by the *csp* operon is the virulence factor induced by temperature changes [114,115] which was involved in bacterial evasion [107]. The identification of CspA sheds light on its involvement in the regulation of bile resistance in *Enterococci faecalis*.

Another MALDI-TOF-MS-based proteomic study of *E. faecalis* reported the upregulation of proteins involved in biofilm formation such as LutC, RsmH, and RRF protein and the downregulation of RepN, ScpA, PrsA, and PurM after antibiotic treatment indicating a decrease in proteins associated with cell division and metabolism during biofilm formation [116].

A nano-LC/MSE (at elevated energy)/(Q/TOF-MS) study reported the upregulation of protein related to glycolysis, amino acid biosynthesis, and biofilm formation. Besides the basic survival pathways, LuxS-mediated quorum sensing, arginine metabolism, rhamnose biosynthesis, and pheromone- and adhesion-associated proteins were upregulated during biofilm formation [117]. Various oxidative stress response proteins and transcriptional regulators correlating with oxidative stress are involved in the pathogenesis of enterococcal infections [107]. The most significant was the identification of peroxide regulator PerR as a ferric uptake regulator-like protein involved in iron homeostasis and OhrR, a transcriptional repressor that senses oxidants [107]. This provides insights into oxidative sensitive targets in *E. faecalis* death with antimicrobial drug intervention.

In *Enterococci,* the expression of Opu, the osmoprotectant uptake transport system, correlates to better survival of bacteria and confers responses to heat shock or other stress factors [118]. This shows that the salt-stress adaptation of *E. faecalis* rather than general stress protection contributes to *E. faecalis* resistance.

The MDR in *Enterococci* is acquired by plasmid pCF10 with pheromone-inducible genes that mediate adhesion and virulence functions through surface proteins amongst which PrgA, B, and C are the main contributor. PrgB is an aggregation factor in biofilm development and virulence enhancement, whereas PrgA is required for *Enterococci* in order to bind to abiotic surfaces, and PrgC’s presence facilitates PrgA function [119].

Raman spectroscopy was also used to analyze the interaction between vancomycin and vancomycin-sensitive *Enterococcus faecalis* strains within a span of 90 min. The effect of the drug was evident from characteristic spectral changes visualized and analyzed with a multivariate statistical model that predicted the impact of vancomycin treatment. The robustness was evident from classification accuracies of >90% at lower concentrations of vancomycin. The Raman spectroscopy methods characterized the drug–pathogen interactions in a label-free and fast method [81]. Vancomycin sensitivity could be noted on the basis of spectral changes with accuracies >90% marking it as a potential tool in diagnosis.

The cellular changes in *E. faecalis* alter the central metabolism and membrane permeability at a low pH. The integration of quantitative proteomic data with a genomic model from SWATH-MS was useful to contextualize these proteomic data [120]. This finding suggests that *E. faecalis* survival is reduced at alkalinity by the blockage of the proton pump.

### 6.2. Staphylococcus aureus

*Staphylococcus aureus* is a Gram-positive cocci, a normal human flora inhabitant, and a nosocomial and community-associated pathogen, causing diverse infections ranging from superficial skin and soft tissue infections to life-threatening infections [121]. Methicillin-resistant *Staphylococcus aureus* (MRSA) strain infections are associated with high morbidity and mortality. Therefore, identifying MRSA is important in targeted hospital infection control measures and the detection of outbreaks [122]. *Staphylococcus aureus* protein A (Spa) typing, multilocus sequence typing (MLST), and pulsed-field gel electrophoresis (PFGE) [123] are the commonly used methods in the detection.

From a proteomics approach, the MALDI-TOF-based MRSA typing scheme has differentiated the major MRSA clonal complexes. It validated a hospital-acquired MRSA (HA-MRSA) typing scheme requiring an average of 2.5 h compared to 3–6 days for the PFGE typing method [124]. The whole-cell MALDI-TOF is also useful in MRSA strain typing [125,126].

An LC-MS/MS study based on iTRAQ reported changes both in the upregulation and downregulation of proteins involved in antimicrobial resistance, stress response, mismatch repair, and cell-wall synthesis. The immunodominant antigen B (IsaB) protein for binding [127] was upregulated in MRSA compared to MSSA. The upregulation of cell-wall-associated fibronectin-binding protein Ebh (for ECM-binding protein homologue) complements resistance in MRSA by altering cell size [128].

However, in one MALDI-TOF-MS-based study, the MRSA and MSSA strains failed to identify a reproducible diagnostic peak but yielded a high discriminative peak with the deployment of artificial intelligence [129,130]. In situations of limited sample availability, the coupling of MALDI-TOF with PBP2a latex agglutination offers a solution for the MRSA assay [131].

The Raman approach discriminated MRSA and MSSA strains in an SCRS at 532 nm excitation and achieved 87.5% accuracy in differentiation. Excitation directly on the bacterial colonies at 785 nm differentiated MRSA and MSSA based on prominent staphyloxanthin bands. A high-intensity band is noted in MRSA strains compared to MSSA, although staphyloxanthin is not linked to antimicrobial resistance mechanisms [132]. The direct application of Raman spectroscopy on bacterial colonies grown on a Mueller–Hinton agar plate yielded 100% accuracy in MRSA detection confirming its potential use in routine clinical diagnostics [133].

The cell size and biochemical features of *Staphylococcus aureus* pose several challenges in their detection. The antibiotic effect signature by SCRS analysis in three cefoxitin-resistant *Staphylococcus aureus* strains and two susceptible strains revealed a weaker *Staphylococcus aureus* spectrum than that previously detected with bacteria–drug combinations and was highly variable. The phenotype correlated with the spectra confirming SCRS can be extended to *Staphylococcus aureus* and introduced into the diagnostic system [134].

Raman microspectroscopy, where Raman spectrometry is coupled to a microscope, had the ability to distinguish between *Streptococcus agalactiae* and *Staphylococcus aureus*. Isogenic variants of *Staphylococcus aureus* strains lacking or expressing antibiotic resistance determinants were also identified and marked as spectral biomarkers. Raman microspectroscopy has the ability to distinguish distinct forms of a single bacterial species *in situ* and thus in detecting antibiotic-resistant strains of bacteria [132].

In addition, a SWATH-based quantitative study in combination with scanning electron microscopy (SEM) and transmission electron microscopy (TEM) validated resistance mechanisms in MRSA [135].

### 6.3. Klebsiella pneumoniae

*Klebsiella* are Gram-negative, encapsulated, non-motile, rod-shaped, and oxidase-negative bacteria [136] classified under the *Enterobacteriaceae* family with a wide diversity of species, including *K. pneumoniae* and others—*K. indica*, *K. terrigena*, *K. spallanzanii*, *K. huaxiensis*, *K. oxytoca*, *K. grimontii*, *K. pasteurii*, and *K. michiganensis*. *K. pneumoniae* accounts for both community- and hospital-acquired infections [137]. *K. pneumoniae* are resistant to third-generation cephalosporins and ESBL strains and are susceptible to carbapenems but account for significant mortality and morbidity [138] as well for the dramatic surge of pan resistance of *Klebsiella pneumoniae* [139,140]. However, recent studies have shown emerging resistance to carbapenems [141].

A high-throughput mass spectrometric analysis of ESBL strains and non-ESBL strains of *Klebsiella pneumoniae* unraveled the pathogenicity determinants. The proteomic analysis identified fimbrial adhesins type 1 and type 3 related to cell invasion [142] and that type 1 fimbrial adhesive proteins facilitate adherence and biofilm formation on abiotic surfaces [143]. The detection of these adhesive structures has paved the way for the development of alternative non-antibiotic strategies targeting the adhesive factors. A shotgun proteomic analysis identified a capsule assembly of Wzi family protein and a capsule in *Klebsiella pneumoniae*, which are critical in bacterial resistance [144] and induce the capacity of the bacteria to enter the bloodstream causing bacteremia and pneumonia in the host [145].

The study by Enany et al., 2020, identified with nano LC-MS different stress response proteins such as the ElaB protein, Lon protease, and universal stress proteins G and A. ESBL strains exhibited unique stress proteins—oxidative stress defense proteins and EntB proteins—with isochorismatase activity, whereas non-ESBL strains had general stress proteins. These proteins facilitate the bacteria to acquire iron and adapt to variable ranges of oxygen levels, for example, hypoxia in the human colon, microoxia at different sites, and hyperoxia in external media. The exploitation of siderophores by bacteria in exhibiting resistance has led to siderophore–drug conjugates and synthetic analogues with therapeutic potential in treatment.

Other unique proteins solely identified in the ESBL-producing *Klebsiella pneumoniae* proteome were the OsmC and general stress protein. OsmC has a critical role in peroxide metabolism and against oxidative stress [146] and general stress protein in the stress resistance response [147].

Clinical studies show that carbapenem-resistant *Klebsiella pneumoniae* (CRKP) account for 70–90% of carbapenem-resistant *Enterobacteriaceae* (CRE) and usually are multidrug-resistant (MDR) [148] with a mortality rate > 50% even after appropriate antibiotic treatment [141]. Colistin is the “last resort” for CRKP infections, and the suboptimal use of it has given rise to colistin-resistant CRKP which are extensively drug-resistant (XDR) strains [149]. A TMT-labeled proteomic technique on both MDR and XDR strains identified DEPs related to drug resistance namely ArnT, ArnD, ArnA, ArnC, ArnB, PmrD, YddW, and OmpK36 in both strains. Notable among them were four β-lactamases, namely, KPC-2, CTX-M-14, SHV-11, and TEM-1, in all the resistant strains. A distinct upregulation of efflux pumps—KexD and AcrA—was noted. The enrichment of WecH, Bm3R1, OppC, OppA, and OppF had the same DEPs in the MDR and XDR strains.

The colistin-resistant XDR strains have a robust biofilm-forming ability and are more resistant [150]. Defects in porins OmpK35 and OmpK36 reduce sensitivity to carbapenems [151,152]. Proteomic analysis detected decreases in the expression of OmpK36 in XDR strains and OmpN in colistin-resistant XDR strains, and the sensitivity to several antibiotics was enhanced with the overexpression of OmpN [153]. The DEPs between the MDR and XDR strains were mainly enriched in cationic antimicrobial peptide (CAMP) resistance and the two-component system—PhoP/PhoQ and PmrA/PmrB [154]. In the CAMP resistance pathway, ArnBCADT, PmrD, and YddW were highly expressed in the colistin-resistant XDR strains, which indicated that lipid A modification persisted as the primary mechanism of colistin resistance in *Klebsiella pneumoniae*.

The two-component system comprises a sensor kinase and a response regulator that maintains bacterial homeostasis including nutrition and antibiotic exposure [155]. The proteomic analysis identified KdpB, OmpK36, PfeA, NasR, NarJ, and ArnB in the two-component system pathway with KdpB being a subunit of K+ transporting ATPase. Among these, Omp 36 is a porin protein important for iron homeostasis [156], PfeA is a ferric enterobactin receptor [157], NasR is a regulator of nitrate/nitrite respiration and assimilation [158], and NarJ is a system-specific chaperone for the respiratory nitrate reductase complex [159].

A comparative proteomic study introduced MICs of a single antibiotic and revealed the role of nutrient modulation in reducing resistance in single-antibiotic-resistant *Klebsiella pneumoniae* [160,161] with a total of nine metabolic pathway proteins (Gar K, UxaC, ExuT, HpaB, FhuA, KPN_01492, FumA, HisC, AroE) being differentially expressed. Similarly, a comprehensive investigation of the proteomes of polymyxin-resistant and polymyxin-susceptible strains of *Klebsiella pneumoniae* revealed that bacterial metabolism plays a crucial role in mediating resistance. For example, the upregulation of the arginine biosynthesis flux after colistin treatment increases the arginine-biosynthetic enzymes ArgABCDE, ArgI, ArgG, and ArgH in colistin-treated *Acinetobacter baumannii* [162] and in gentamicin-treated *Staphylococcus aureus* [163]. Arginine metabolism in *Klebsiella pneumoniae* moderates hydroxyl-radical-induced damage via ammonia production [164].

One study reported the impact of colistin in decreasing the expression of the maltose transporter LamB, a porin involved in the influx of antibiotics and the class A β lactamases—TEM, SHV-11, and SHV-4 [165]. Comparative proteomic analysis of polymyxin-susceptible *Klebsiella pneumoniae* validated the role of *crrB*-mediated colistin resistance in which lipid A profiles presented the addition of one or two L-Ara4N molecules and palmitoylation with elevations in CrrAB, PmrAB, and ArnBCADT levels. The multidrug efflux pump KexD and the GNAT family N-acetyltransferase were highly expressed in the crrB mutant. Thus, the proteomic study confirmed the role of *crrB* mutation in colistin resistance [164].

UV resonance Raman (UVRR) spectroscopy applied for the differentiation of *Klebsiella pneumoniae* outperformed Raman microspectroscopy with 92% accuracy in species classification [166].

### 6.4. Acinetobacter baumannii

*Acinetobacter baumannii* are Gram-negative, round, rod-shaped bacteria (coccobacillus) that predominantly cause nosocomial infections primarily, such as ventilator-associated pneumonia (VAP) [21,167]. Carbapenem-resistant *Acinetobacter baumannii* (CRAB) is ranked as a number-one-priority organism by the WHO. For multidrug-resistant strains of *Acinetobacter baumannii* (MDR-AB), carbapenem is the preferred treatment drug [168]. However, prior use of carbapenem increases resistance to carbapenem [169]. The alternate treatment options for MDR-AB are polymyxins [170,171]. *Acinetobacter baumannii* strains resistant to three or more classes of antimicrobials (penicillins and cephalosporins—including inhibitor combinations, fluoroquinolones, aminoglycosides, and carbapenems) are classified as extensive drug-resistant strains (XDR-AB), and XDR-AB strains resistant to polymyxins and tigecycline are pandrug-resistant (PDR-AB) [172,173].

In the MDR strain, the upregulation of antibiotic-resistant proteins β-lactamases (AmpC, Oxa-23 carbapenemase, and TEM), outer membrane proteins (OmpA, a CarO homolog, OmpW, NlpE homolog involved in copper resistance), drug-modifying enzymes (aminoglycoside acetyltransferases, aminoglycoside 3′, phosphotransferase, nitroreductase DrgA), and drug transporters (a homolog of the ABC transporter HlyD; the AcrB-AdelJK cation/multidrug efflux pump) were noted. Host defense proteins, CRISPR-associated proteins (Csy3 and Csy1), LexA-like regulator (SOS response), and cell surface porin DcaP-like protein for biofilm formation, have been noted.

A TMT labeling and label-free proteomic study identified metal-dependent hydrolase-related proteins and β-lactamase-related proteins upregulated in MDR strains. Aminoglycoside-modifying AphA1b was uniquely expressed in MDR strains. Antibiotic-resistant protein DacD (D-alanyl-D-alanine carboxypeptidase), a PBP6b, and cell division protein ZapA, involved in β-lactam resistance, were also upregulated. The ABC transporter, MFS transporter, and RND transporter were upregulated. Stress-response-related proteins—Trigger factor (TF), Heavy-metal-associated (HMA), Rhodanese-Like Domain (RHD), Universal stress protein (Usp), AldA, and CysK—were upregulated. 

A 2D-DIGE, MALDI-TOF/TOF, and iTRAQ/SCX-LC-MS/MS study identified the unique biofilm capability of *Acinetobacter baumannii* [174]. A 2DE and LC-MS/MS study noted the overexpression of proteins involved in iron storage, the metabolic process, and lipid biosynthesis while an iron-deficient condition leads to the overexpression of proteins involved in iron acquisition [175]. Quantitative phosphoproteomics identified the phosphorylation sites in *Acinetobacter baumannii* by LTQ-Orbitrap MS enriched by SCX-TiO2 chromatography [176].

A comparison of the spectral difference in *Acinetobacter* strains by Raman spectroscopy emphasizes its advantages and the rapidity of the discriminative power compared to MS. Further, the performance of Raman spectroscopy was superior in *Acinetobacter baumannii* strain differentiation as it contained whole-cell information [177].

### 6.5. Pseudomonas aeruginosa

*Pseudomonas aeruginosa* are Gram-negative, aerobic–facultatively anaerobic, rod-shaped bacteria that frequently establish bacteremia in neutropenic patients causing high morbidity and mortality rates [178,179]. It is a model organism in understanding biofilm physiology and antibiotic tolerance. It is the primary causative organism of chronic infections in chronic cystic fibrotic lungs by forming biofilms that are refractory to the host immune system and antimicrobial therapies [180]. *Pseudomonas aeruginosa* accounts for >5% of infectious exacerbations in chronic obstructive pulmonary disease (COPD) patients and associated mortality [181].

The resistance mechanisms exhibited by *Pseudomonas aeruginosa* are intrinsic, acquired, and adaptive. Intrinsic resistance results from low outer membrane permeability and expression of the efflux pump. It acquires resistance either by horizontal gene transfer or from mutations in resistance genes [182]. Adaptive resistance is marked by the formation of a biofilm that serves as a diffusion barrier [183]. In addition, multidrug-tolerant cells form a biofilm as is the case with cystic fibrosis patients [184].

Virulence factors are not expressed constitutively but are cell-density-dependent and sensed by a diffusible molecule such as N-acyl homoserine lactone (AHL), in a process known as quorum sensing [185,186,187]. In *Pseudomonas aeruginosa*, quorum sensing is regulated by the *las* and *rhl* system that is interrelated. *las* mediates transcriptional activator LasR and LasI and an AHL synthase to synthesize N-3-oxo-dodecanoyl-homoserine lactone (3-oxo-C12-HSL). The *rhl* system mediates RhlR and RhlI for the synthesis of N-butanoyl homoserine lactone (C4-HSL). The *las* system is an activator of rhlR and rhlI. Mutations in the quorum sensing circuitry lower virulence [188,189,190]. Proteomic analysis of post-translational modifications in *Pseudomonas aeruginosa* PAO1 quorum-sensing (QS) system revealed differentially expressed proteins partly rescued only by a medium containing AHL signal molecules [191]. Another study also revealed that the inactivation of the QS system termed “quorum quenching” results in the reduced expression of many extracellular virulence factors, including proteases, chitinase, and lipases [192,193] and the downregulation of the type II Xcp secretion system [194]. The outer membrane hemin-binding receptor PhuR was positively regulated by AHL, demonstrating that the *has* system (haem acquisition system) and the Phu Haem acquisition system are regulated by the *las*I *rhl*I QS circuitry.

The LC-ESI MS/MS study identified DEPs that correspond with porins OprD, OprE, OprF, OprH, and Opr86, LPS assembly protein, and A-type flagellin. Significant downregulation of flagellin A protein, OprF, and OprD and the upregulation and modification of OprH, OprE, Opr86, and LptD are noted in tolerant strains reflecting the adaptability of bacteria in conditions in which porins play an important role.

Proteomic studies by iTRAQ revealed the involvement of biofilm formation in antibiotic resistance mediated by proteins ArcA and IscU. Antibiotic resistance alterations by drugs also showed changes in the expression of the proteins PhzA, PhzB, PhzM, MetQ1, ArcA, IscU, lpsJ, and PilA involved individually or synergistically in the regulation of PA quorum sensing, the bacterial secretion system, bacterial biofilm formation, and CAMP resistance [195].

Detection of carbapenemases activity is challenging which has been simplified by a modified MALDI-TOF MS assay that detects the β-lactam ring and its degradation products. Β-lactamases disrupt the central β-lactam ring of drugs by hydrolysis, and this hydrolysis corresponds to a mass shift of +18 Da that is easily detected by MALDI-TOF MS. This method has validated β-lactamase activity in *Acinetobacter baumannii* [196]. In the case of assays involving meropenem, the visualization of degradation products by MALDI-TOF MS is difficult due to their binding to cell lysate components. The modified method detects degradation products and has been validated with NDM-1-, VIM-1-, KPC-2-, KPC-3-, and OXA-48/-162-producing members of the *Enterobacteriaceae* and NDM-1-producing *Acinetobacter baumannii* isolates [197,198,199].

By convention, carbapenemase strains are identified by phenotypic methods such as the modified Hodge test. Carbapenems in combination with different inhibitors (e.g., cloxacillin, EDTA, or 3-aminophenyl boronic acid (APB)) are used to differentiate among AmpC, metallo-β-lactamases (MBLs), and *Klebsiella pneumoniae* carbapenemase (KPC). The MBLs are identified by inhibition with EDTA, for differentiating between MBL and other carbapenemases in *Enterobacteriaceae* and *Pseudomonas* spp. in a MALDI-TOF platform [200].

*Pseudomonas aeruginosa* adapt to low-oxygen environments, and the protein involved in this adaptation was investigated by both SWATH MS and data-dependent SPS-MS3 of TMT-labeled peptides. Under hypoxic stress (O_2_ < 1%), both aerobic (Cbb3-1 and Cbb3-2 terminal oxidases) and anaerobic denitrification and arginine fermentation proteins were increased [201]. Another proteomic analysis using iTRAQ technology identified DEPs associated with resistance mechanisms such as quorum sensing, bacterial biofilm formation, and active pumping [195].

### 6.6. Enterobacter spp.

*Enterobacter* are Gram-negative, rod-shaped bacteria in the *Enterobacteriaceae* family. *Enterobacter aerogenes* and *Enterobacter cloacae* are clinically significant species that are opportunistic, nosocomial pathogens originating from intensive care units especially on mechanical ventilation [202]. Colistin, a cationic lipopeptide, is administered to treat multidrug-resistant (MDR) *Enterobacter* infections [203], including ESBL strains and/or resistant to carbapenems [173]. Cell membrane electronegativity is lowered by modifying lipid A, which decreases the binding affinity of colistin [204]. The classical methods of testing colistin susceptibility are challenging [205], due to the lack of reproducibility, inconsistencies [206], and limitations and due to inaccurate MICs resulting from the adherence of colistin to the testing wells [207]. Moreover, limitations exist in the protein-based MALDI-TOF MS detection of *Enterobacter* infections and a modified lipid-based MS platform [208].

The lipid-based MS is the fast lipid analysis technique (FLAT) on a MALDI-TOF/MS platform that rapidly identifies Gram-negative and Gram-positive bacteria [209,210]. FLAT-MS is a highly sensitive method in identifying CRE and *K. aerogenes* [211]. The modifications in the terminal phosphates of lipid A with phosphoethanolamine, L-amino-4-arabinose (Ara4N), or galactosamine confer colistin resistance [212] but are detected by MALDI-TOF.

MALDI-TOF clustering confirmed the existence of a preferential way of transmission for Gram-negative bacteria from the invasive procedure employed. LC-MS/MS identified potentially pathogenic factor OmpX as the most abundant protein in *Enterobacter cloacae* OMVs with hydrolase enzymes, that cause cell interaction [213] and enhance immune tolerance [214] and the passage of microbial molecules through the tight junction of the gut [215]. The OMVs assist in the formation of biofilm, as indicated by the presence of OmpX.

Finally, a specific robust method to comprehensively detect ESKAPE pathogens at a single-cell level uses Raman microspectroscopy. The spectral features were distinct for each of the pathogenic bacteria and thus facilitated the identification [216]. Raman scattering microscopy was also useful for the rapid identification and AST of pathogens in urine [217] and notable in its ability to classify on a Gram-staining basis and AST results within ~3 h drawing attention for clinical applications.

## 7. Summary and Perspective on the Role of Proteomics in Microbial Resistance Diagnosis

A number of proteomic tools have been used in the detection of AMR for ESKAPE pathogens. To be practical and useful in the routine practice of clinical microbiology labs, the proteomic tool should be accurate, rapid, and cost-effective. Modern microbiology has attempted to introduce technology into laboratories that includes MALDI-TOF MS “profiling” or “biotyping” as the first-line identification method as it involves a very simple sample preparation. The workflow illustrates the direct smearing of a bacterial sample onto the MALDI target, a short chemical extraction classically after overnight cultivation, covered by a simple layer of one of the standard matrices, which is followed by the acquisition of a suitable number of profile spectra from randomly chosen locations of the sample spot. In “fingerprinting”, the peak list extracted from an averaged profile spectrum is compared to the reference spectra peak lists. This has developed into a routine tool for microbial identification transforming clinical microbiology. The rapid success of MALDI-TOF-MS is attributable to the accuracy of identification, speed of analysis enabling earlier implementation of therapy, and significant cost effectiveness, thus outperforming earlier clinical routine tests based on biochemical reactions. Further, it has excellent performance data on the accuracy of identification including those difficult to analyze by traditional methods as in the case of Gram-negative non-fermenting bacteria.

MALDI-TOF profiling has comparative accuracy comparable to DNA-based methods. For instance, in cystic fibrotic patients, biotyping for the identification of Gram-negative, non-fermenting bacilli improves treatment outcomes, as they are life-threatening organisms. It has also been useful in identifying anaerobic bacteria which are generally difficult to identify by traditional clinical microbiological methods such as *Clostridium*, *Bacteroides*, *Prevotella*, etc. A simple MALDI-TOF profiling approach has a chance of identifying bacteria that are rare and difficult to culture and highly pathogenic bacteria, such as *Francisella tularensis*, *Brucella* spp., *Burkholderia mallei*, and *Burkholderia pseudomallei*.

Prior inactivation has to be applied to highly pathogenic microorganisms before analyzing them in a MALDI-TOF mass spectrometer to prevent any contamination of the instrument and avoid health risks for the users. A comparison with partial 16S rRNA gene sequencing for difficult-to-analyze bacteria revealed correct identification (85.9%) in the MALDI-TOF MS profiling, with the misidentification resulting from laboratory errors rather than the failure of method.

The MALDI-TOF-MS-based species identification of bacteria provides results reproducible within 10 min without any substantial costs for consumables. The MALDI protocol is able to identify 1.45 days earlier on average. Incorporation of the MALDI protocol significantly reduces reagent and labor costs together with a remarkable decrease in waste disposal as well. In ESKAPE pathogens, proteomic antibiotic resistance detection has been noted predominantly involving MALDI-based technologies.

The proteomic profiling of ESKAPE pathogens involving various technologies and their relevance to antibiotic resistance is summarized in Table 1.

## 8. Pros and Cons of Proteomics in AMR

Antibiotic resistance is a serious problem. Proteomic studies in their detection has provided vital information as it provides the entire protein profile after exposure of the resistant, intermediate, and susceptible bacteria to sublethal antibiotic concentrations. The response to antibiotics involves proteins related to almost the entire metabolic processes such as energy, nitrogen metabolism, nucleic acid synthesis, glucan biosynthesis, and stress response. Usually, proteomic expression profiles are confirmed with a genomic and/or transcriptome analysis including post-transcriptional modifications. The major pros are that the proteomic tools are more functionally and phenotypically relevant than genetic/genomic assays. In the era of massive nucleic acid sequencing, proteomic tools are promising to mitigate the gap between nucleic acid sequencing and AMR. The advantage of providing more phenotypically relevant information in AMR is crucial because there is still a considerable discrepancy between nucleic acid sequencing and AMR. Harnessing proteomic tools in AMR is needed in the investigation of AMR.

The major cons are that most proteomic tools are expensive, labor-intensive, and time-consuming. While the cost of nucleic acid sequencing has dropped significantly over the past two years and will continue to drop in the near future, the cost of proteomics tools does not appear to be decreasing in the near term. More importantly, most proteomic tools are labor-intensive and lack automation. This would have a significant impact on their widespread use in clinical microbiology while addressing the massive clinical testing demands. Due to a lot of manual processes, proteomic tools are relatively time-consuming. When the flaws are not significantly improved, proteomic tools can be used only for research but are not possible to be widely used in clinical microbiology.

Proteomic tools are more functionally and phenotypically relevant than genetic/genomic assays. Among the proteomic tools, MALDI-TOF MS is only one proteomic tool that is rapid and accurate. Thus, easy sample preparation and short turn-around time make MALDI-TOF a practical tool that has been widely used in clinical microbiology labs. Some studies have reported successful AMR detection based on the MALDI-TOF MS spectra. However, the wide application of MALDI-TOF MS in AMR detection has not met a general agreement yet. The reason could be that results reported from various countries and teams were still significantly different.

The difference in prevailing local strains can be the possible mechanism explaining the performance discordance. One of the possible solutions is using a more sophisticated AI algorithm to build a more robust MALDI-TOF AI model, so the AI model can be generalized to every local region. By contrast, another solution is to train a locally useful MALDI-TOF AI model based on locally relevant MS data. In the methodology, the idea of one-fits-all generalization is abandoned. Instead, a locally tailored MALDI-TOF AI model is the focus. Further investigations addressing the issue are still on the way.

## 9. Conclusions

In conclusion, proteomics plays a critical role in providing functionally relevant information in the study of bacterial resistance diagnostics. From simple two-dimensional gel electrophoresis to mass spectrometry, current proteomics methods used for microbial studies are reliable. With the combined capabilities of top-down and bottom-up approaches, proteomics can pursue studies ranging from the quantification of gene expression to host–pathogen interactions. As evidenced by the recent pandemic, it is noteworthy that proteomic advances can aid in the diagnosis of ESKAPE resistance and prevent the next impending pandemic of antibiotic resistance. Moreover, the cost of proteomic techniques is effective when considering the laborious bacterial culture techniques. Together with genomics, advances in proteomic tools promise to provide a more comprehensive view of antibiotic resistance mechanisms and diagnostics.

## Figures and Tables

**Figure 1 diagnostics-13-01014-f001:**
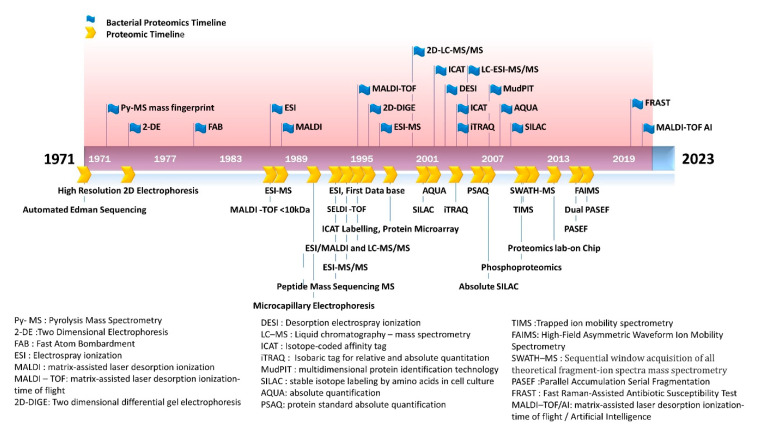
Emergence of bacterial proteomics in line with proteomics milestone. The figure shows the application of proteomic tools in bacterial testing.

**Figure 2 diagnostics-13-01014-f002:**
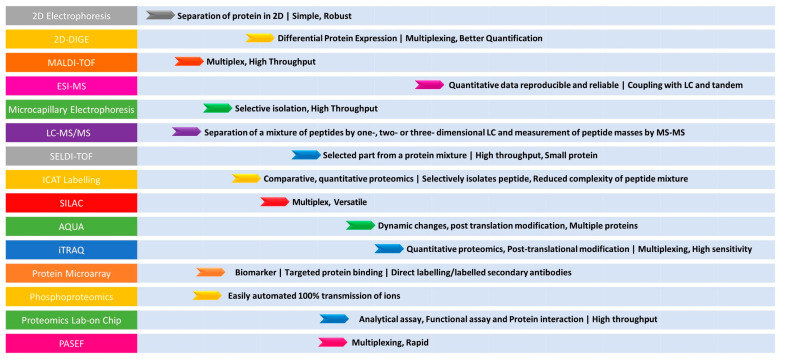
Summary of the advantages for various proteomic tools in AMR detection. The (**left**) panel indicates the techniques. The (**right**) panel indicates the application and its advantage.

**Table 1 diagnostics-13-01014-t001:** Proteomic techniques for ESKAPE resistance.

Pathogen	Proteomic Technique	Physiological Effect	Reference
*Enterococcus*	2DE/MALDI-TOF/MS	Resistance—VanA, VanB,	[107]
		Heat shock response—CspA	
	LC-MS	Peptidoglycan synthesis— d-Ala-d-Ala	[218]
	TIMS-TOF	Multidrug resistance—EfrA, EfrB	[219]
	Nano-LC MS	OptrA protein, Esp protein	[220]
		Surface exclusion protein—Sea1	
		Conjugal transfer protein—TraB	
		Replication protein—RepA	
		XRE—transcription regulator protein	
	MALDI-TOF	Typing VanB positive	[116,221]
	MS/MS	LPxTG—Ace, Acm, Scm	[222]
		Pili—Ebp, PilA, PilB	
	iTRAQ	Biofilm formation—strong and weak biofilm forming	[223]
	Raman spectroscopy	Vancomycin resistance detection	[81]
	SWATH-MS	PFL, LDH1	[120]
*Staphylococcus aureus*	MALDI-TOF	PBP2a	[129,224]
	Tandem MS	β-lactam resistance, BORSA, MODSA	[225,226]
	SWATH-MS	MRSA mechanism	[135]
	Raman spectroscopy	Coagulase strain identification	[227,228]
	MALDI-TOF/MS	Typing MRSA vs. MSSA	[125,229,230]
	LC-MS	Endogenous peptides for differentiation	[231]
	2DE	Alkali shock protein 23—Asp23,	[232]
		Cold-shock protein—CspABC	
		Virulence regulator—SarA	
	iTRAQ-LC-MS/MS iTRAQ/MS	Ftsh, AtpA, AtpC, AtpD, AtpH, GlyA	[233,234]
		β-lactam resistance—PBP2′,	
		bifunctional autolysin—Atl, FmtA, PBP2,	
		peptidoglycan elongation protein MurA2,	
		transglycosylase domain protein—Mgt, teicoplanin resistance TcaA,	
		LCP domain-containing proteins—MsrR	
*Klebsiella* *pneumoniae*	1D-LC MS/MS	Porins—LamB, CirA, FepA, OmpC	[235]
	iTRAQ/LC-MS/MS	Colistin resistance—CrrAB, PmrAB, PhoPQ,	[164]
		ArnBCADT, PagP	
		Multidrug efflux pump—KexD	
	iTRAQ	Capsule production proteins—Wza, Wzb, Wzc, Wzi, Gnd, Ugd, Wca, CpsB, CpsG, GalF in ESBL+	[236]
		TreA, Wza, Gnd, RmlA, RmlC, RmlD, GalE, AceE, SucD Porins—OmpK35, OmpK36	
	LC-MS	Carbapenemase activity	[237]
	MALDI-TOF	Differentiates carbapenemase vs. metallo β lactamases	[238,239,240]
		Carbapenemase	
		Carbapenemases—KPC-1, GES-5, NDM-1, VIM-1, VIM-2, IMP-1, GIM-1, SPM-1, OXA-48, OXA-162	
	SILAC	CRKP outer membrane	[241]
	Raman Spectroscopy	Differentiate *K. pneumoniae* strains	[166]
*Acinetobacter baumannii*	2DE/MS-MS 1D/LC/MS-MS	Antibiotic stress proteins— OmpA_38_, CarO, OmpW	[242]
	2DE	AmpC, Cpn60 chaperonin, ATP synthase, OmpA	[243]
	2DIGE	Omp A, CarO, CsuA/B	[244]
		Inner membrane fraction	[245]
	TMT-LC-MS	Β-lactamase—Oxa23	[246]
	MALDI-TOF	MDR—biotyping	[247]
		Carbapenemase detection	[248,249]
	Raman spectroscopy	Epidemiological analysis	[250]
	MALDI-TOF/MS	MDR proteins	[251]
		Quorum sensing—AHL	[252]
	iTRAQ	OmpW	[253,254]
	TRAQ/SCX-LC-MS/MS	Biofilm—CsuABABCDE chaperone	[174]
*Pseudomonas* *aeruginosa*	MALDI-TOF	Quorum sensing	[255]
		Antibiotic resistance proteins—OprG, OprF, MexA, OprD, OmpH	[256]
	MALDI-TOF/MS	Metallo β lactamases	[238,257]
	LC-ESI MS/MS	OprE, OprH, Opr86	[258]
	BONCAT	Biofilm	[70]
	2DGE/MALDI TOF	Quorum sensing protein—PhuR, HasAp	[259]
	2DGE/XCT MS	Adaptive resistance—porins (OprF and OprG) and lipoproteins (OprL and OprI)	[260]
	SWATH-MS	Cbb3-1, Cbb3-2 terminal oxidases	[201]
		NarG, NarH, NarI nitrate oxidases	
		ArcA, ArcB, ArcC, PchA-G, FpvA, FpvB, FptA, PhuR, HasR, PutA, KatG, KatE, Dps	
	Raman Spectroscopy	Quorum sensing	[261]
	iTRAQ	Biofilm—ArcA, IscU	[195]
*Enterobacter* spp.	DIGE/LC-MS/MS	ESBL	[262]
	LC-MS	OMPV	[263]
	MALDI-TOF-MS	MDR—carbapenem resistance	[264]

## Data Availability

Data sharing not applicable.

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
