# Peer review of "Antibiotic Resistance Diagnosis in ESKAPE Pathogens—A Review on Proteomic Perspective"

_diagnostics, 2023, doi:10.3390/diagnostics13061014_

Round 1
Reviewer 1 Report
Dear authors
It is a very interesting topic.
The text needs some editing.
It is better to refer to alternative ways for antibiotics such as phage therapy.
Reviewer 2 Report
I congratulate the authors for a well-documented and realized work.
I have a single observation regarding compliance with the instructions for authors, especially regarding references.
Reviewer 3 Report
The review is exciting, well described and a vital report on applying proteomics to detect AMR, particularly for ESKAPE pathogens.
1- The writing of genes and some microorganisms should be revised. For example, Enterococci faecalis to be Enterococcus faecalis, and the names of genes must be in italics. spp. is not written in italic
2- The manuscript should be revised thoroughly for English.
3- A section on The pros and cons of protein-based analysis methods in detecting antimicrobial resistance mechanisms should be added.
